# Climate-trait relationships exhibit strong habitat specificity in plant communities across Europe

Stephan Kambach [1,2] ✉, Francesco Maria Sabatini [1,2,3,4], Fabio Attorre [5], Idoia Biurrun[6], Gerhard Boenisch[7], Gianmaria Bonari [8], Andraž Čarni [9,10], Maria Laura Carranza [11], Alessandro Chiarucci [3], Milan Chytrý [12], Jürgen Dengler [13,14], Emmanuel Garbolino [15], Valentin Golub[16], Behlül Güler[17], Ute Jandt [1,2], Jan Jansen[18], Anni Jašková[12], Borja Jiménez-Alfaro[19], Dirk Nikolaus Karger [20], Jens Kattge [2,7], Ilona Knollová[12], Gabriele Midolo[12], Jesper Erenskjold Moeslund [21], Remigiusz Pielech [22], Valerijus Rašomavičius [23], Solvita Rūsiņa [24], Jozef Šibík [25], Zvjezdana Stančić [26], Angela Stanisci[11], Jens-Christian Svenning [27], Sergey Yamalov[28], Niklaus E. Zimmermann [20] & Helge Bruelheide [1,2]

Ecological theory predicts close relationships between macroclimate and functional traits. Yet, global climatic gradients correlate only weakly with the trait composition of local plant communities, suggesting that important factors have been ignored. Here, we investigate the consistency of climate-trait relationships for plant communities in European habitats. Assuming that local factors are better accounted for in more narrowly defined habitats, we assigned > 300,000 vegetation plots to hierarchically classified habitats and modelled the effects of climate on the community-weighted means of four key functional traits using generalized additive models. We found that the predictive power of climate increased from broadly to narrowly defined habitats for specific leaf area and root length, but not for plant height and seed mass. Although macroclimate generally predicted the distribution of all traits, its effects varied, with habitat-specificity increasing toward more narrowly defined habitats. We conclude that macroclimate is an important determinant of terrestrial plant communities, but future predictions of climatic effects must consider how habitats are defined.

Predicting the effects of a changing climate on the diversity and functioning of the ecosphere requires an understanding of how climate drives the distribution of plant species and ecosystem properties[1,2]. Ecosystem functioning, such as productivity and nutrient cycling, is strongly determined by the functional composition of the plant community[3–6]. Functional traits represent species' life-history strategies[7], and are often summarized with a few main, largely independent, axes of variation, such as the fast-slow continuum[8], as reflected in the leaf-economics spectrum[9], the species' reproductive strategy[10], the plant size spectrum[7], and the continuum of collaboration with mycorrhizal fungi[11]. A foundational, yet globally weakly supported, assumption in trait-based ecology is that the geographical distribution of dominant functional traits in plant communities is shaped by macro-environmental gradients, independently of taxonomy[12–14]. Here, we addressed this assumption by studying the consistency of macroclimate-trait relationships among European plant communities.

The most popular approach to studying climate-trait relationships involves regression analysis between climatic gradients and the community-weighted mean of plant functional traits (CWMs)[15]. At the global scale, however, CWMs have been weakly predicted by linear gradients of climatic conditions[13]. Yet, in Australia, which can be considered a relatively homogeneous evolutionary unit, climate predicted 43% of the variation in CWMs[16], suggesting that global climate-trait relationships might be blurred by different phylogenetic histories when modelled across biogeographic realms. Furthermore, these relationships could also be significantly affected by habitat-specific factors[17]. When considering local plant communities, CWMs are likely shaped by soil conditions[18,19], microclimate[20], human land use[21], disturbance[22], biotic interactions[23] (e.g. herbivory or competition[24]) and evolutionary history that constrainsthe available species pool[25]. All of these factors likely interact with macroclimatic gradients. Hence, they may mask the actual effects of climate on local plant communities.

The classification of plant communities into hierarchically arranged habitat types (hereafter habitats) offers a promising approach to studying climate-trait relationships because it accounts for interfering local factors that define different habitats. Restricting analyses to habitats with similar floristic composition, shared evolutionary history and comparable environmental conditions could disentangle the effects of interacting factors and result in tighter climate-trait relationships, as has been observed in analyses focused on specific habitats (e.g. forest and tundra habitats[20,26]). Yet, it is unclear whether the strength and expression of climate-trait relationships depend on the thematic resolution (hereafter "narrowness") of the habitat definition. Analysing climate-trait relationships at different levels of habitat narrowness could help us understand whether these relationships are consistent across both levels (resolution-invariant) and habitats (i.e., habitat-specific). This understanding might be important for more accurate predictions of the effects of a changing climate on the distribution and functioning of natural plant communities.

We used >300,000 geo-referenced vegetation plots (Supplementary Fig. S1) from the European Vegetation Archive (EVA)[27], 19 high-resolution bioclimatic variables from the CHELSA database[28,29] and species-level plant trait information from the TRY database[30,31]. Each plot was characterized by the CWM of four key functional traits: plant height, specific leaf area (SLA), seed mass and specific root length (SRL) (Table 1). Previous research has shown that these four traits can capture the major gradients in the global spectrum of plant form and function (plant height, SLA and seed mass)[7] and in the root economic spectrum (SRL)[11]. Each plot was classified to three hierarchical levels of the EUNIS habitat classification[32,33] which include eight broadly defined level 1 habitats (hereafter broad habitats), 40 more narrowly defined level 2 habitats (intermediate habitats) and 216 of the most narrowly defined level 3 habitats (narrow habitats)[33]. We applied generalized additive mixed-effects models that accounted for spatial dependence among plots to determine the linear effects of the principal components of the European climatic gradients (climate PCs) on the distribution of the four CWMs. We hypothesized that (i) climate has a general effect on CWMs across habitats (consistent with previous findings, see Table 1), but (ii) classifying plots into more narrowly defined habitats increases the proportion of CWM variation that can be explained by climate. Furthermore, we examined whether narrowly defined habitats exhibit climate-trait relationships similar to those observed in the more broadly defined habitats. We show that the observed effects of macroclimate on the distribution of plant functional traits are sensitive to the type and narrowness of the plant communities studied.

## Results

### Climate-trait relationships across broad habitats
We summarized the information from the 19 bioclimatic variables into four principal components (PCs) using principal component analysis.

These four gradients jointly captured 88.4% of the variation in macroclimatic conditions (Supplementary Fig. S1) and represented the European gradients from subarctic to dry-summer Mediterranean climates (PC1, hereafter Mediterranean gradient), from colder continental to warmer coastal climates (PC2, temperature gradient), from colder to warmer summer climates (PC3, summer-temperature gradient) and a gradient of increasing seasonality of precipitation and temperature regimes (PC4, seasonality gradient). All four climatic gradients were significant predictors of the four CWMs across all broad habitats (Fig. 1 and Supplementary Fig. S2). An increase in the Mediterranean gradient was associated with higher community-weighted plant height, seed mass and SRL and lower SLA. An increase in temperature gradient was also associated with higher community-weighted plant height, seed mass and SRL and lower SLA. The positions of the broad and intermediate habitat types along the four climatic PCs are shown in Supplementary Figs. S3–S6.

### Climate-trait relationships along the hierarchy of habitats
Within the broad habitats, we observed several deviations from the general climate-traits relationships observed across habitats. Along the Mediterranean gradient, community plant height decreased in forest and coastal habitats; SLA increased in wetlands and SRL decreased in coastal saltmarshes, forests, and man-made habitats. Along the temperature gradient, plant height decreased in coastal habitats; SLA increased in four of the eight broad habitats and SRL decreased in heathland and forest habitats. The effects of climate on community seed mass were generally more consistent than for the other three traits (Fig. 1 and Supplementary Fig. S2).

Habitat was generally a better predictor of CWMs than the combined fixed and habitat-specific effects of climate. In models that did not account for the habitat-specificity of climate-trait relationships (Cl +broad in Fig. 2), community plant height was the trait best explained by broad habitats (with a maximum of 77.2% of explained variation), followed by seed mass (67.1%), SLA (53.3%) and SRL (45.6%). When accounting for habitat-specificity in narrow habitats (see Cl*narrow in Fig. 2), these values increased to 92% for plant height, 73.9% for seed mass, 63.7% for SLA, and 53.6% of the variation in SRL. Descending the classification hierarchy towards more narrowly defined habitats, we found that the proportion of variation in CWMs explained by climatic conditions increased for two of the four plant traits (from Cl*broad to Cl*narrow in Fig. 2). For SLA and SRL, the proportion of explained variation increased toward more narrowly defined habitats (with a maximum of 10.3% and 8.0% of explained variance, respectively) whereas plant height and seed mass showed no such pattern (with a maximum of 5.7% and 6.8% of explained variance, respectively). For plant height, SLA and seed mass, we found that climate-related variation in CWMs was maximized when habitat type was not accounted for (see Cl in Fig. 2). Nevertheless, the habitat-specific interaction terms in our models explained a significant proportion of CWM variability for all traits and all levels of the classification hierarchy (Supplementary Table S1).

### Climate-trait relationships in intermediate and narrow habitats
In the intermediate-level habitats, the observed climate-trait relationships mostly matched those in the more broadly defined habitats, albeit with an increased proportion of nonsignificant and sometimes contrasting relationships (see Fig. 3 for the exemplary relationships of the first and second principal components with plant height, Supplementary Fig. S7 for the third and four principal components and Supplementary Figs. S8–S13 for the other traits with the four principal components). Within narrow habitats, we observed that climate-trait relationships were frequently habitat-specific, often with contrasting relationships among habitats that were part of the same superior broad habitat (Fig. 4). This habitat-specificity of climate-trait relationships was observed for all traits and in all broad habitats.

## Table 1 | Description of the analysed functional traits

| Trait | Ecological function | Reported relationships with precipitation | Reported relationships with temperature | References |
|---|---|---|---|---|
| Plant height | Central to species' carbon allocation strategy and ability to compete for light. | Positive | Positive | 16,53,67 |
| Specific leaf area (SLA) = ratio of leaf area to dry mass | Central to species' acquisition strategy along the leaf economic spectrum. | Nonsignificant or positive | Negative or positive | 9,16,37,53 |
| Seed mass | Determines dispersal and seedling survival under unfavourable conditions. | Nonsignificant or positive | Nonsignificant or positive | 53,68–70 |
| Specific root length (SRL) = ratio of root length to dry mass | Central to species' collaborative strategy as species with mycorrhizal associations tend to have lower SRL (thicker roots). | Nonsignificant or positive | Negative | 39,71,72 |

Referenced studies were conducted across large climatic gradients and multiple habitats.

## Discussion

Understanding whether climate-trait relationships are habitat-specific or general, and whether these relationships are invariant to the narrowness of the applied habitat definition could improve our predictions of how plant communities will be affected by future climate change. In this study, we summarized the effects of climate on four key functional plant traits—plant height, SLA, seed mass and SRL—from broadly to more narrowly defined habitats in Europe. While climate was found to be a significant predictor of CWMs for all four plant traits, both within and across broad habitats, our results revealed three key findings. First, habitat type was generally a better predictor of trait distributions than climate per se. Yet, for plant height, SLA and seed mass, climate and habitat shared some of the explained variation in trait distribution, suggesting that climate also strongly influences the broad distribution of habitat types (see Supplementary Figs. S3–S6). Second, decomposing the analysis into increasingly narrow habitats revealed stronger climate-trait relationships for SLA and SRL but not for plant height and seed mass. Third, in the most narrowly defined habitats, the effects of climate on the expression of all four traits were strongly habitat-specific, regardless of which broad habitat type or climatic gradient was considered. By using increasingly narrow habitat definitions, we were able to reduce the effects of local factors on CWMs and thus better unveil the effects of macroclimate on the functional trait composition of plant communities.

The habitat classification we used is based on floristic and biogeographic characteristics. As a result, plant communities were analysed within homogeneous groups that reflect common adaptations to local environmental features that are either independent of global climatic gradients[34] or difficult to capture with global datasets on climate and soil conditions. We found that the effects of climate on plant community composition in broad habitats are not necessarily reflected in similar effects in more narrowly defined habitats[35]. For instance, community-weighted mean plant height increased overall in grassland habitats along the Mediterranean gradient, but this pattern reversed when focusing on seasonally wet and wet grasslands (habitat R3). Similarly, we found that higher temperatures led to higher community plant height in wetland habitats, except for periodically exposed shores (Q6), where temperature was negatively related to plant height. Our results on the habitat-specificity of climatic effects are in accordance with previous findings on the distinct effects of precipitation on vegetation productivity in grassland versus forest habitats in China[36]. On the global scale, the predictive power of climate for the distribution of community leaf trait composition increased when 14 rather than four broad habitat types were considered[37]. This finding provides an explanation for why studies in more narrowly defined habitats revealed more pronounced effects of climate on plant traits (e.g. as shown in tundra habitats[26]) than studies across habitats[13]. The two community traits that were less well predicted by habitat alone (namely SLA and SRL) were increasingly better predicted by climate when habitats were more narrowly defined. The underlying reason might be that SLA depends more strongly on soil than on climatic conditions[19,38] and that the relationship between SRL and environmental gradients also strongly depends on the habitat considered[39]. Taken together, these observations suggest that broad habitats include species with different life-history strategies (in terms of competition for light and collaboration with mycorrhizal fungi), while more narrowly defined habitats include species with more similar life-history strategies. Consequently, broad habitats might allow for a wider range of alternative climatic responses, while in narrow habitats, species tend to share similar climatic responses. Thus, the effect of climate on the proportion of species with high or low SLA (or SRL) becomes clearer. The two traits that did not follow this pattern, i.e. community plant height and seed mass, are strongly correlated within the global spectrum of plant form and function[7,40] and most strongly determined by habitat type. Community plant height is related to the occurrence of

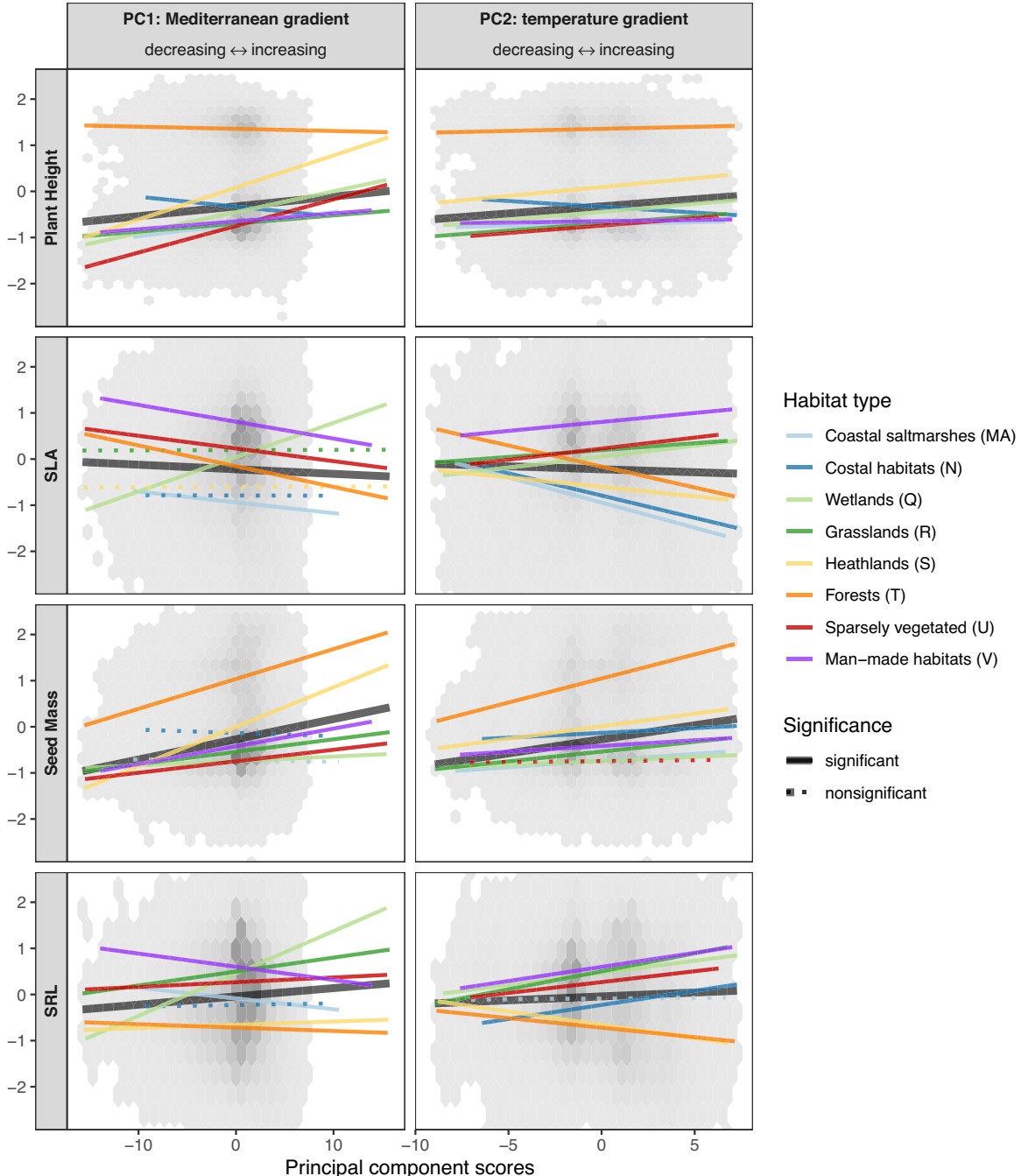

**Fig. 1 | Effects of climate on four plant traits in broad habitats.** The graphs show the community-weighted means of four plant functional traits as linear functions of the first and second principal components (PCs) of the 19 CHELSA bioclimatic variables, obtained with generalized additive mixed-effects models. Slopes show the relationships across all (black) and within the most broadly defined habitats of the EUNIS classification (colours). Solid lines indicate significant relationships at $p < 0.05$ (based on separate two-sided $t$ tests). Grey hexagons show the distribution of plot-level observations. SLA specific leaf area, SRL specific root length.

woody species, which markedly depends on land use and is also more strongly determined by soil moisture and nutrients than by broad climatic conditions[41].

Although the investigated climate-trait relationships were mostly consistent across broadly defined habitats, we found several habitat-specific relationships. For instance, the increase in plant height along the Mediterranean and temperature gradients could be explained by the longer growing season and higher light availability, which both increase competition for light in warmer regions[42,43]. In forest habitats, the decrease in plant height along the Mediterranean gradient could be attributed to the limiting effects of higher

temperatures and lower water availability on the growth of canopy trees. The distribution of forest understory traits, however, is fairly independent of precipitation gradients in Europe[20]. Community height in coastal habitats could be limited by precipitation because dune soils are highly permeable (and thus edaphically dry) and the groundwater is not available to most plants[44]. For SLA, we observed a decline along the Mediterranean gradient (i.e., SLA decreased with increasing drought stress), corresponding to the global[13] and intra-specific reduction of SLA under drought[45]. The same arguments apply to the effects of temperature. Wetland communities, for instance, might be able to maintain a higher SLA, even under higher temperatures,

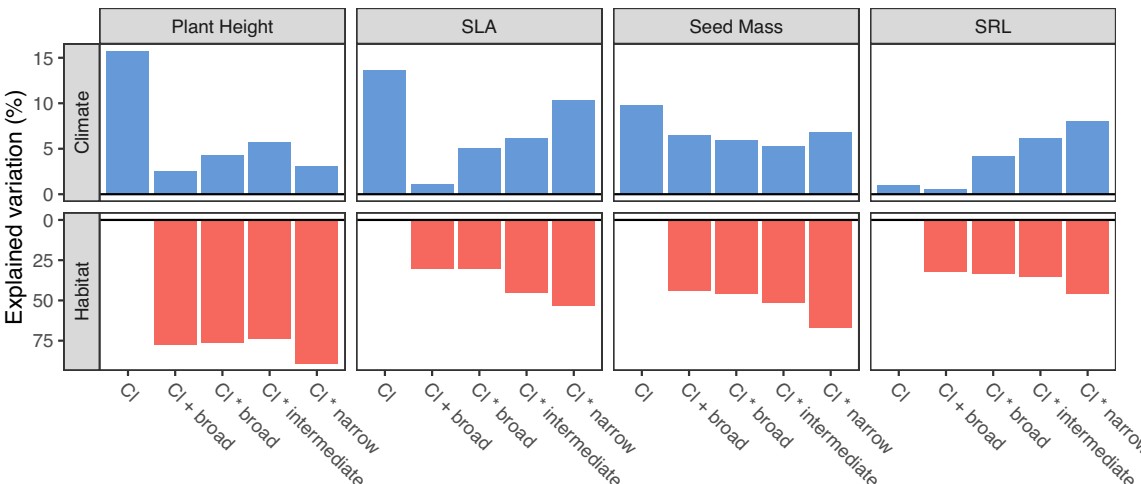

**Fig. 2 | Proportion of variation in plant traits explained by climate and increasingly narrow habitat definitions.** The graphs show marginal R² values from generalized additive mixed-effects models for the linear dependence of the community-weighted means of four plant functional traits on habitat (red bars) and the general and habitat-specific effects of climate (blue bars). Model complexity increases from left to right. Cl: effects of climate across all plots or habitats, modelled with fixed effects of the four principal components of the 19 bioclimatic variables. CL + broad: fixed effects of climate plus random intercept effects of broad habitats. Cl*broad/intermediate/narrow: fixed effects of climate plus the random effects of broad, intermediate, or narrowly defined habitats plus the habitat-specific effects of climate, modelled with random-slope effects between the climatic principal components and the respective habitats. SLA specific leaf area, SRL specific root length.

because they are not constrained by water availability. Grassland species might benefit from higher SLA in warmer temperatures if they are able to compensate for natural or human-induced disturbances by regrowing relatively quickly. In man-made habitats, the response of SLA might depend on the relative abundance of different life forms, with therophytes being relatively more abundant at higher temperatures[46]. Community seed mass showed the greatest consistency in climate-trait relationships across all broad habitats. Yet, the increase in seed mass along the Mediterranean gradient was quite unexpected, as seed mass tends to increase with precipitation and decrease with aridity on the global scale[13] or is only weakly related to precipitation in Mongolia, China[47] and Australia[48]. In habitats where water availability depends on groundwater, such as coastal salt marshes, coastal habitats and wetlands, community seed mass was independent of the Mediterranean gradient. An explanation might be that larger seeds, which can buffer the survival of seedlings and saplings[49], are most important during hot and dry summers. For community SRL, the observed increase along the Mediterranean and temperature gradients refuted our expectations. In coastal saltmarshes, heathland and forest habitats, however, we also observed decreases in SRL along both gradients, suggesting that the effects of macroclimate may be strongly habitat-specific or masked by local factors, such as local soil conditions[19]. These local conditions were increasingly accounted for by including more narrowly defined habitats in the models of climate-SRL relationships.

In order to synthesize the effects of macroclimate across a large-scale dataset, we had to accept some shortcomings in our analysis. We focused only on the linear effects of climate and ignored interactions among the different dimensions of climate, although we can assume that the distribution of the four plant traits is also shaped by nonlinear and interaction effects (c.f. tree height[50] and leaf traits[20]). We partially accounted for non-linearity by using log-transformed CWMs. Still, our models identified stronger linear effects of climate on community traits compared to global-scale analyses[13]. Whether the classification into narrower habitats might have strengthened or alleviated any nonlinear or interaction effects has, to our knowledge, not been tested yet. By focusing exclusively on climate, we did not account for important interactions with edaphic conditions (such as evapotranspiration), which are the best predictor for leaf area and maximum

plant height distributions across Australia[16]. Furthermore, it is conceivable that the effects of climate on community trait composition could change at the end of climatic gradients. This assumption could be tested by comparing trait-environment relationships under benign versus extreme conditions[39]. In forests, we did not differentiate between the canopy and understory layers, although these layers might show contrasting responses to climate[20]. Finally, we calculated CWMs based on species-level trait averages rather than in situ measurements, which precluded any analysis of intra-specific trait variation. These shortcomings may have diluted the effects of climate on CWMs, but were unavoidable given the large geographic extent of our study.

Our results show that macroclimate is a consistent predictor of the CWMs of plant traits in all habitats and across all climatic gradients in Europe. These effects of climate vary across habitats, depending on the thematic resolution of the habitat definition and the identity of the habitat. We thus anticipate that climate-change effects will not be limited to particularly sensitive communities[51], but will have diverse effects on a wide range of plant communities[52]. To accurately predict the effects of a changing climate on the composition and functioning of the ecosphere, we recommend considering the habitat-specificity of climate-trait relationships.

## Methods
### Vegetation data
The raw vegetation survey data consisted of 1,741,856 plots with 37,318,600 species records from 107 databases, collated and curated by the European Vegetation Archive (EVA)[27] and accessed on May 12th, 2021. Individual databases are listed in Supplementary Data 1. Species abundances were measured or converted to percentage cover. Before the analysis, we removed all plots (1) with only presence/absence data, (2) without geographic coordinates, (3) situated in Greenland, (4) with latitudinal coordinates lower than the southernmost point of 34° N or higher than the northernmost point of 82° N, (5) with trait information for less than 80% of the total plant cover and (6) for which climatic variables could not be calculated due to empty pixels around the focal plot. To reduce the effects of spatially clustered and repeated censuses, we only retained the most recent census for each unique location and randomly stratified the dataset to include only one plot per

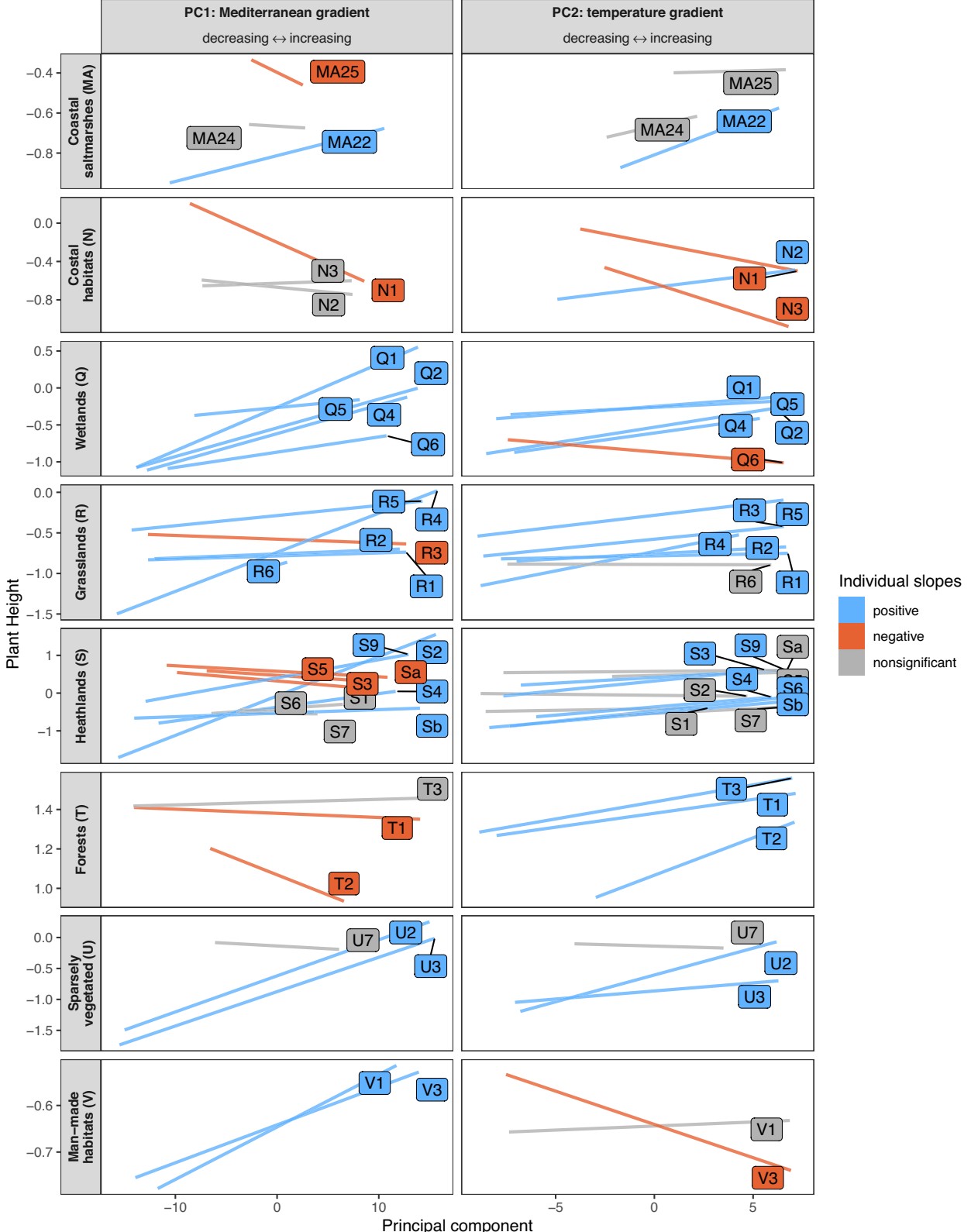

**Fig. 3 | Effects of climate on plant height in intermediate-level habitats.** The graphs show the community-weighted means of plant height as a linear function of the first and second principal components (PCs) of the 19 CHELSA bioclimatic variables, obtained from generalized additive mixed-effects models. Significance was determined at $p < 0.05$ (based on separate two-sided $t$ tests). Slope estimates from habitats with fewer than 100 plot observations were omitted. Codes within boxes refer to habitat types (as listed in Supplementary Data 2).

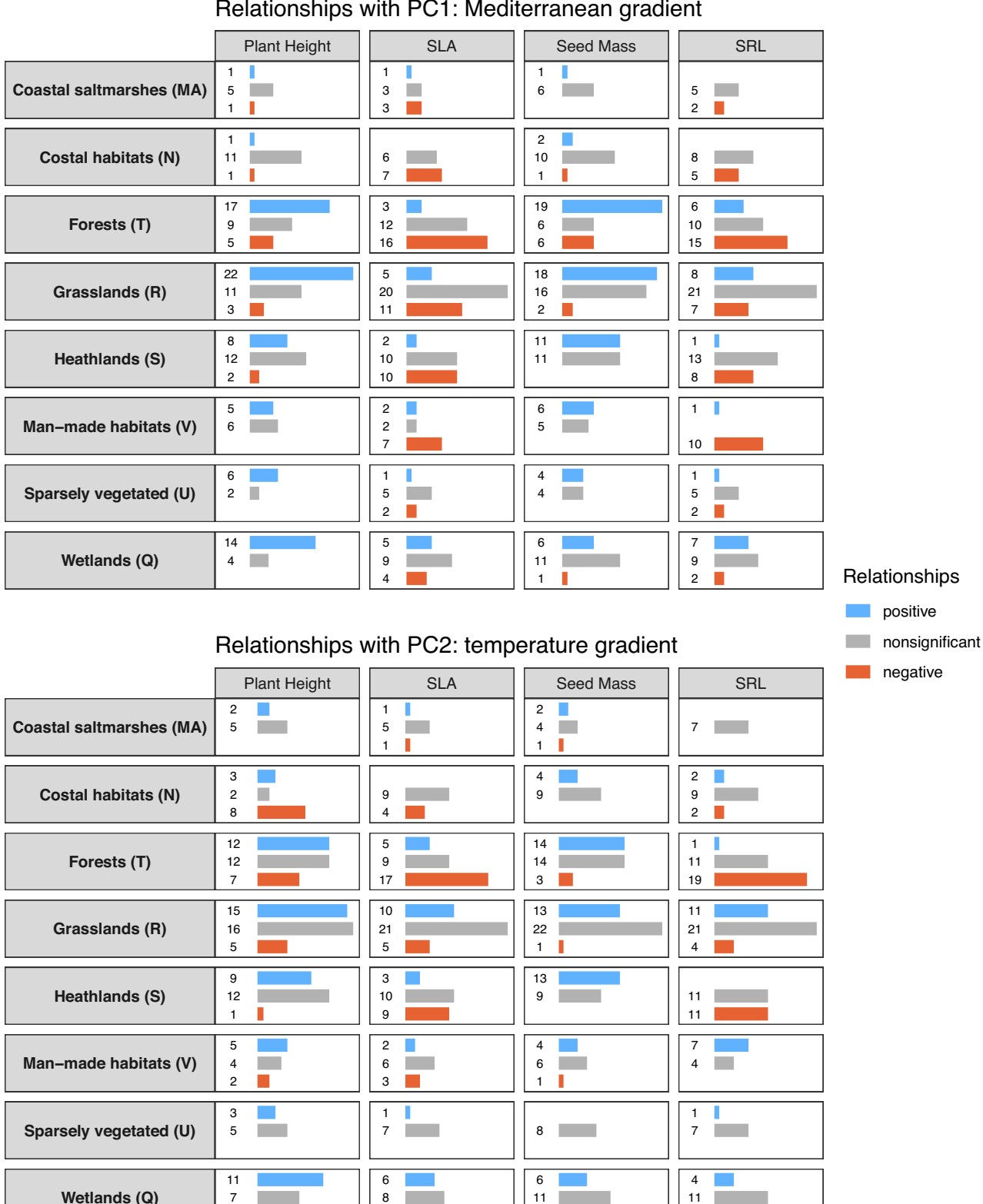

**Fig. 4 | Effects of climate on plant traits in narrowly defined habitats.** The graphs show the number of slope estimates between the first and second principal components (PCs) of the 19 CHELSA bioclimatic variables and the community-weighted mean of four functional traits, calculated with generalized additive mixed-effects models. Colouring indicates the expression and significance of the relationship. Significance was determined at $p < 0.05$ (based on separate two-sided $t$ tests). Slope estimates from habitats with fewer than 100 plot observations were omitted. SLA specific leaf area, SRL specific root length.

0.01° grid cell (approximately 1.11 × 0.69 km in Central Europe). The resulting dataset contained 300,021 vegetation plots with 7,320,239 individual species occurrence observations that were conducted between 1873 and 2020 and distributed between 34.8° N to 80° N latitude and 10° W to 59.3° E longitude with a focus in Central Europe (Supplementary Fig. S1).

## Taxonomy

We harmonized species names from the European Vegetation Archive using the Taxonomic Name Resolution Service 5.0 (tnrs.biendata.org). Subspecies and varieties were merged at the species level. Species names were matched to the taxonomic backbone 3.0 of the sPlot global vegetation database[53] to maximize compliance with species names from the TRY plant trait database[30]. Fungi, bryophytes, algae, and lichens were removed from the data, resulting in 13,758 vascular plant species.

## Community-weighted trait means

We extracted species- and genus-level average values for plant height, SLA, seed mass and SRL from the TRY plant trait database[30] version 5, which covered species-level mean values of 33 traits for 50,404 species, derived with Bayesian Hierarchical Probabilistic Matrix Factorization. This algorithm predicts individual-level trait values based on the observed trait records, observed trait-trait correlations, and the taxonomic hierarchy reflecting phylogeny[54,55]. The root-mean-square error of the z-transformed predicted versus available trait values ranged from 0.07 (seed mass) to 0.2 (SLA) (see Supplementary Fig. S14). From this gap-filled dataset, we assigned species-level mean trait values to 6,025 species from our dataset and to an additional 690 species, we assigned genus-level mean trait values from the gap-filled dataset. The relationships between the four focal traits and 24 traits from the gap-filled dataset are shown in Supplementary Fig. S15. The original publications for the data used are listed in Supplementary Table S2. Without imputed values, we were able to assign trait values to 45% (plant height), 28.4% (SLA), 34.4% (seed mass) and 5.4% (SRL) of all species in the dataset used for the analysis, which accounted for an average of 95.6%, 91.4%, 90.8% and 47.4% of plant cover at the plot level, respectively. For each plot, we calculated the CWM of the four plant traits according to the equation in[13].

## Habitat classification

Each plot was assigned to a habitat type based on the classification expert system EUNIS-ESy (European Nature Information System[32,33]), updated on October 25th, 2021. We used the first three levels of the classification hierarchy, ranging from the highest level 1 (broad habitat types, e.g. forests−habitat code T) through the intermediate level 2 (intermediate types, e.g. broadleaved deciduous forests – T1) to the lowest level 3 (narrow types, e.g. temperate *Salix* and *Populus* riparian forest – T11). This classification system included the following broadly defined habitat types at level 1: coastal salt marshes (MA, 3,085 plots), coastal habitats (N, 5,045 plots), wetlands (Q, 28,931 plots), grasslands (R, including lands dominated by forbs, mosses or lichens, 107,457 plots), heathlands (S, including shrubland, scrub and tundra, 25,866 plots), forests (T, including other wooded land, 98,311 plots), inland sparsely vegetated habitats (U, 2,932 plots) and vegetated man-made habitats (V, 28,394 plots). All intermediate and narrow habitats are listed in Supplementary Data 2. Factsheets on the floristic composition of EUNIS habitats are listed in[33] and[56].

## Climatic data

For each plot, we extracted the 19 bioclimatic variables from the CHELSA Climatologies version 1.2[28,29] at a pixel resolution of 30 arc sec (~1 km) using the cloud-based Google Earth Engine platform. Bioclimatic values were calculated as the average value from all pixels within 500 m from the plot coordinates. Plots with missing climate data in adjacent pixels were omitted from further analyses. Covariation among the 19 bioclimatic variables was removed by a principal component analysis. The first four principal components (PCs) jointly explained 88.4% of the climatic variation (Supplementary Fig. S1). We used the loadings of the 19 bioclimatic variables to extract the positions of all plots along these four PCs, which were used as predictors in the subsequent analyses.

## Statistical analysis

Prior to the following analyses, all plot-level CWMs were log transformed to approximate normality. We estimated the effects of the four climatic PCs on the CWMs of the four traits using separate generalized additive mixed-effects models that always included a spline-on-the-sphere smoothing term (based on latitude and longitude[57]) to account for spatial dependence among plots[58]. For each of the four traits, we created the following five different models: Model 1 included only the fixed-effects of the four climatic PCs without any interactions. Only in this model, we accounted for the different number of observations per habitat type by including only the same number of randomly selected plots from each of the broad habitats (based on the minimum number of 2,932 plots in U – Sparsely vegetated habitats). Model 2 included the fixed-effects of the four climatic PCs plus the random effects of the broad level 1 habitat types. Model 3 included the fixed-effects of the four climatic PCs, the random effects of the broad level 1 habitat types plus random interaction terms between the fixed effects of the climatic PCs and the broad level 1 habitat types. Models 4–5 were similar to model 3 except that the broad level 1 habitat types and their random interaction terms were replaced with the intermediate level 2 or the narrow level 3 habitat types and their respective interactions with the climatic PCs. The implementation of all models is documented on GitHub. Model residuals were visually checked and are shown in Supplementary Figs. S16-S20.

For models 3–5, we quantified the effects of climate on trait CWMs, both across and within habitats, with predicted marginal mean regression slopes. All regression slopes were assigned an approximate confidence interval (slope estimate ± 1.96 * standard error) and confidence intervals that did not include zero were considered significant at approximately $p < 0.05$. We did not test regression slopes that were based on fewer than 100 plot observations. For each model, we calculated the proportion of variation in CWMs that could be explained by climatic gradients (fixed effects of climatic PCs and interactions between climatic PCs and habitat types) and habitat type (random effects). The proportion of explained variation was quantified using the marginal and conditional $R^2$[59], whose calculation for random-slope models was based on the variance components of the fixed and random effects and whose implementation we adopted from the R script provided by[60]. To check whether the obtained results depended on the proportion of species with actual trait measurements, we repeated all analyses with models in which the contribution of each plot was weighted according to the summed cover of species with trait values divided by the total plant cover. The results of these weighted analyses were qualitatively similar to the results of the unweighted analyses and are shown in Figs. Supplementary S21–S25. The code implemented to calculate the marginal and conditional $R^2$ for generalized additive mixed-effects models is documented on GitHub.

All analyses were conducted in $R$[61] using the package *mgcv* for generalized additive mixed-effects models[57,58], *emmeans* and *ggeffects* for marginal mean regression slopes[62,63], *dismo* for geographical stratification of plots[64] and *ggplot2* for data visualization[65].

## Reporting summary

Further information on research design is available in the Nature Portfolio Reporting Summary linked to this article.

## Data availability

The data generated in this study (plot-level information on plot coordinates, survey year, CWMs and climatic PCs) were deposited in the data repository of the German Centre for Integrated Biodiversity Research (iDiv) Halle-Jena-Leipzig (https://doi.org/10.25829/idiv.3527-g89efk). Raw vegetation data are available under restricted access because they belong to the owners of each vegetation database, but can be requested at euroveg.org/eva-database-obtaining-data. The bioclimatic and plant trait data used in this study can be downloaded from chelsa-climate.org/bioclim and www.try-db.org/TryWeb/dp.php, respectively.

## Code availability

The code used to conduct the following steps of the analyses is available at github.com[66] (https://doi.org/10.5281/zenodo.7404176): (i) extracting the plot-level bioclimatic variables, (ii) running and analysing the generalized additive mixed-effects models, (iii) creating the presented figures.

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

## Acknowledgements

The 2019-2020 BiodivERsA joint call for research proposals, under the BiodivClim ERA-Net COFUND program, and with the funding organisations Swiss National Science Foundation SNF (project: FeedBaCks, 193907), Agence nationale de la recherche (ANR-20-EBI5-0001-05), the German Research Foundation (DFG BR 1698/21-1, DFG HI 1538/16-1), and the Technology Agency of the Czech Republic (SS70010002) funded this research and the position of S.K. and G.M. The Basque Government (IT1487-22) supported the work of I.B. The Slovenian Research Agency (ARRS P1-0236) supported the work of A.Č. The University of Latvia (grant nr. AAP2016/B041//Zd2016/AZ03) supported the work of S.R. Financial support within the Rita-Levi Montalcini (2019) programme, funded by the Italian Ministry of University, supported the work of F.M.S. The Slovak Research and Development Agency (grant nr. APVV 16-0431) supported the work of J.Š. The VILLUM FONDEN (grant 16549) funded the VILLUM Investigator project "Biodiversity Dynamics in a Changing World" which supported the work of J.C.S. J.C.S. also considers this work a contribution to Center for Ecological Dynamics in a Novel Biosphere (ECONOVO), funded by Danish National Research Foundation (grant DNRF173). S.K. acknowledges financial support from the Open Access Publication Fund of the Martin Luther University Halle-Wittenberg and the iDiv Open Science Publication Fund.

## Author contributions

S.K., H.B., F.M.S. designed the research. S.K. analysed the data. F.A., I.B., G.Bon., A.Č., M.L.C., A.C., M.C., J.D., E.G., V.G., B.G., U.J., J.J., A.J., B.J.-A., M.L., J.E.M., R.P., V.R., S.R., J.Š., Z.S., A.S. and J.-C.S. contributed the data. G.Boe., J.K. and I.K. curated the data. M.C. assigned the plots to the EUNIS classification. N.E.Z. coordinated the Feed-BaCks project and acquired funding. S.K. wrote the manuscript with substantial contributions from H.B. and F.M.S. and contributions from all co-authors.

## Funding

## Competing interests

The authors declare no competing interests.

## Additional information

[1]Institute of Biology/Geobotany and Botanical Garden, Martin Luther University Halle-Wittenberg, Halle, Germany. [2]German Centre for Integrative Biodiversity Research (iDiv), Halle-Jena-Leipzig, Leipzig, Germany. [3]BIOME Lab, Department of Biological, Geological and Environmental Sciences (BiGeA), Alma Mater Studiorum University of Bologna, Bologna, Italy. [4]Faculty of Forestry and Wood Sciences, Czech University of Life Sciences Prague, Prague – Suchdol, Czech Republic. [5]Department of Environmental Biology, Sapienza University of Rome, Roma, Italy. [6]Department of Plant Biology and Ecology, Faculty of Science and Technology, University of the Basque Country UPV/EHU, Bilbao, Spain. [7]Max Planck Institute for Biogeochemistry, Jena, Germany. [8]Faculty of Science and Technology, Free University of Bozen-Bolzano, Bolzano, Italy. [9]Research Centre of the Slovenian Academy of Sciences and Arts, Jovan Hadži Institute of Biology, ZRC-SAZU, Ljubljana, Slovenia. [10]University of Nova Gorica, School for Viticulture and Enology, Nova Gorica, Slovenia. [11]Envixlab, Department of Biosciences and Territory, University of Molise, Pesche, Italy. [12]Department of Botany and Zoology, Faculty of Science, Masaryk University, Brno, Czech Republic. [13]Vegetation Ecology Research Group, Institute of Natural Resource Sciences (IUNR), Zurich University of Applied Sciences (ZHAW), Wädenswil, Switzerland. [14]Plant Ecology, Bayreuth Center for Ecology and Environmental Research (BayCEER), University of Bayreuth, Bayreuth, Germany. [15]Climpact Data Science (CDS), Nova Sophia – Regus Nova, Sophia Antipolis Cedex, France. [16]Samara Federal Research Scientific Center, Institute of Ecology of the Volga River Basin, Russian Academy of Sciences, Togliatti, Russia. [17]Biology Education, Dokuz Eylul University, Izmir, Turkey. [18]Department of Ecology and Physiology, Faculty of Science, Radboud University, Nijmegen, the Netherlands. [19]IMIB Biodiversity Research Institute (Univ.Oviedo-CSIC-Princ. Asturias), University of Oviedo, Oviedo, Spain. [20]Swiss Federal Research Institute WSL, Birmensdorf, Switzerland. [21]Section for Biodiversity, Department of Bioscience, Aarhus University, Aarhus, Denmark. [22]Department of Forest Biodiversity, University of Agriculture in Krakow, Kraków, Poland. [23]Institute of Botany, Nature Research Centre, Vilnius, Lithuania. [24]Faculty of Geography and Earth Sciences, University of Latvia, Riga, Latvia. [25]Institute of Botany, Plant Science and Biodiversity Center, Slovak Academy of Sciences, Bratislava, Slovakia. [26]Faculty of Geotechnical Engineering, University of Zagreb, Zagreb, Croatia. [27]Center for Biodiversity Dynamics in a Changing World (BIOCHANGE), Department of Biology, Aarhus University, Aarhus C, Denmark. [28]Botanical Garden-Institute, Ufa Scientific Centre, Russian Academy of Sciences, Ufa, Russia. ✉e-mail: stephan.kambach@gmail.com

