## [Peer Review File · Nature Communications]

Climate-trait relationships exhibit strong habitat specificity in plant communities across EuropeREVIEWER COMMENTS

Reviewer #1 (Remarks to the Author):

Kambach and colleagues tested whether community weighted means (CWMs) in four functional traits (seed mass, plant height, specific leaf area (SLA), specific root length (SRL)) are related to climate variables and whether the explanatory power of the climate variables get stronger if one accounts for habitat type. To this aim, they combined more than 300,000 vegetation records from the European Vegetation Archive, which have been assigned to the hierarchy of EUNIS habitat types, with climatic data from the Chelsa database and with gap-filled trait data from the TRY database. They found that for CWMs of SLA and SRL, the variance explained by climatic variables increased when they used finer habitat categories. They also found that the relationship between CWMs and climate may depend on the habitat type, and they therefore conclude that it is important to account for habitat type.

The manuscript is overall clearly written, but as it is very brief, I am missing some important details on the methods and I am also missing some results or discussion thereof. For example, although for CWMs of SLA and SRL, the amount of variance explained by climate increases when they included finer and finer habitat categories, it is not mentioned that for three of the traits, the explained variance in CWMs is actually highest when habitat is not considered. This makes sense if one considers that habitat types are also determined by climate. It would be useful to discuss this and also to show the positions of the habitat types in PCA plots (or tables).

As the introduction nicely starts with functional traits and main axes of variation, such as the fast-slow continuum, it would be nice if they could point out how the four chosen traits relate to these axes. For example, based on a recent paper by Carmona et al. (2021, Nature), plant height and seed mass would both represent their first axis, SLA their second axis, and SRL their third axis. So, the traits chosen represent three of the major axes of trait variation, and this is a strength of this study.

Of the more than 1.7 million vegetation plots in the European Vegetation Archive, the authors only used the c. 300,000 that had trait data for the species that covered more than 80% of the plots. This is good. However, as the authors used the gap-filled (imputed) trait data of TRY, I wonder how good some of the trait data and thus the values of the CWMs are. For details on the gap-filling, one is referred to another paper, but the authors should mention for how many of the 6,586 species used in the analyses, the data was imputed. In the Carmona et al. study on the global spectrum of plant traits and function, they has SRL data for only 1,736 species. I assume that this number is not higher here, for the European flora, which would mean that measured SRL data would only be available for up to 25% of the species and that the remaining data for SRL have been imputed. This does not mean that the data are not correct, but it would be good to provide the details. If there is large variation in the percentage species/coverage

in CWMs that are based on 'real' and imputed data, one could consider using the percentage of real data per plot as a weighting factor in the analyses.

I would also like to see more details on the actual statistical analyses. The authors used generalized additive models (GAMs). As they also included random terms, I assume that it were generalized additive mixed models (GAMMs). I am not a specialist in GAM(M)s, but I thought that they fitted non-linear functions to the data, but based on the figures, it seems that only linear relationships were fitted. So, how do the used models differ from generalized linear (mixed) models? Also, the method of calculating the marginal R² is based on the paper by Nakagawa and Schielzeth (Ref. 56), which is about GLMMs. Again, this does not mean that the analyses are wrong, but more details on the analyses should be provided.

Minor comments

Line 156: place 'the' before 'more'

Line 168: CWMs, not CMWs

Line 170: insert 'also' before 'associated' to indicate that it is the same pattern as for the Mediterranean gradient.

Line 250-252: Here the authors could expand a bit on why it is likely that plant height and seed mass might be more strongly driven by habitat. To me this seems obvious given that plant height is strongly associated with woodiness, and that therefore e.g. forest habitats have much higher plant heights than grasslands. And seed mass is frequently associated with plant height (both align well along trait-spectrum axis 1 of Carmona et al.

Line 288: The focus on linear effects is mentioned as a limitation. I do not see this as a major limitation, but when the authors used GAMs, I wonder why they did not allow them to be non-linear.

Line 302: Add here that another limitation of the data are that many data were imputed.

Line 325-326: I assume that these latitude cut-offs correspond to the most southern and northern latitudes of Europe, but it would be good to explicitly mention why these cut-offs were chosen.

Line 330: Add the approximate size in km² of the 0.01° grid cells.

Line 347-348: Provide more information on the gap-filling. For how many species per trait was this done, and how good were the gap-filling statistics?

Line 388: Here and elsewhere, the habitat categories start with a capital letter. Initially, this confused me, but then I realized that it is most likely the code for the habitat type. To avoid confusion, I recommend to put delete the code from the text or to put it in brackets after the written-out name of the habitat type: sparsely vegetated habitats (U).

Fig. 1: I assume that the range of x-axis values covered by the lines varies because some of the habitat types did not cover the full range of principal component scores. It would be good to mention this in the caption. It would also be helpful to plot the average or median principal component score of each habitat type, either as a dot on the corresponding line, or as vertical stripes on the x-axis.

Reviewer #2 (Remarks to the Author):

Habitat-specificity of climate-trait relationships

Stephan Kambach and colleagues evaluated relationships between climate and plant functional traits across fine scales in Europe. By calculating community-weighted means of plant height, SLA, seed mass and SRL within vegetation plots, they found increases in predictive power of climate varying from broader to narrow finer scales for SLA and SRL but not height and seed mass. The findings have implications for understanding the drivers of plant distribution especially for predicting the effects of climate on species distributions.

Because of the very narrow focus of the work, I recommend that it may be more suited for a discipline-focused outlet dealing with plant ecology than a general-readership journal like Nature Communications as I find the 'general readership' component, i.e., why a general readership should care, missing. However, I leave this to the editor to decide.

I have the following comments below:

1) It was not clear why the authors focused only on the four plant functional traits: plant height, SLA, seed mass and SRL. The less traits data may give unreal signature, or cannot represent the full nature of climate-trait relationships.

2) In the context of this analysis, functional traits are properties of species. However, species are not independent units--instead, they are descended from shared phylogenetic ancestry. Yet, I could not find anywhere the authors accounted for such phylogenetic autocorrelation or even acknowledge this fact. Such autocorrelation can surely influence the results presented here.

3) While the study dwelled on the scale-dependence of habitat and climate on plant functional traits, it was not clear the size of the different habitat types examined. For instance, the authors used ambiguous terms like 'narrower' or 'broad' habitats, but it is not clear for a reader how big or small these habitats actually are. In Line

Reviewer #3 (Remarks to the Author):

The authors explore climate-trait relationships to test a foundational assumption of trait-based ecology: that “the geographical distribution of dominant functional traits in plant communities is shaped by macro-environmental gradients, independently of taxonomy”. That is to say, the authors want to test whether there is some generality in the trait values geographical distribution according to macro-environmental gradients.

They argue that trait-climate relationships at a global scale are often weak or inconsistent because global climate-traits relationships might be blurred or masked by local conditions, habitat-specific factors, or different evolutionary histories. To understand whether climate-trait relationships are global or habitat specific, and whether these relationships are invariant to habitat narrowness definition, the authors suggest restricting the analysis to smaller units (more homogenous units) like for example habitats that shared floristic composition, or evolutionary history, that reflect common adaptations that are probably independent of global climate gradients. So, they can control these confounding or interacting factors.

Within these habitats they also suggest analyzing climate-trait relationships using different levels of habitat definition by changing the scale, basically (if I am understanding properly what they called “habitat narrowness”). This understanding might improve trait-climate predictions and help to understand the consequences of functional variation on ecosystem functioning.

To answer these questions, the authors employed an impressive vegetation dataset from the European Vegetation Archive, 19 high-resolution CHELSA bioclimatic variables, and they characterized the CWM of four functional traits obtained from TRY plant trait database. The habitat types were classified according to the EUNIS classification, with three levels of narrowness (broad-intermediate-narrow habitats).

They found that climate is a significant predictor of CWMs of all plant traits analyzed in all habitats, which is in line with previous studies. They find that decomposing the analysis into narrower habitats improve climate-trait relationships for some traits (SLA and SRL), but not all. The authors found that trait composition was largely determined by habitat type (that is to say, there are strong habitat-specificity in the functional composition of communities). However, they were able to demonstrate a strong effect of climate on functional composition of communities irrespective of habitat, which from my point of view is their most noteworthy result.

General comments to the study:

I find the work original, well written, easy to follow, the methods are state of art to me, and it also provide evidence on climate-traits relationships. In addition, this work contributes to test a foundational assumption of trait-based ecology, that has been relatively poorly tested and that is key for understanding climate impacts on communities, and ecosystems. I personally appreciate to see this kind of works in the literature since they help to solidify foundational trait-based ecology postulates. I have read the work carefully and did not find any major issues to point out. I would just appreciate to see the distribution of the models' residuals to check whether the model is correctly specified.

Regarding details for reproducibility. I consider the authors provided enough details about the data employed and the methods. However, a reproducible code for harmonization, data cleaning and data analysis is missing. Also, the final dataset is missing (or at least I couldn't find them). In my opinion, at least the data analysis codes and final data set should be available for the readers in a final version of this manuscript.

Minor comments on the methods and data analysis:

I do not find any major issues or flaws in the methods and data analysis. I found the data analysis appropriate to test their hypothesis and the work support the conclusions and claims.

Although already addressed in the text, I would like to point out a few minor concerns.

First, the authors calculate CWM from species-level trait averages. They do not use in-situ/local measurements neither any measure of intraspecific variation in their analysis. They acknowledge this shortcoming and justify the decision due to the large geographic extent of the study what is reasonable to me. Although not needed for this study, it would be awesome to have local measures, or some proxy of the intraspecific trait variation (ITV) in the field to see if there are improvements on climate-trait relationships (Although is still a preprint you can have a look to Maitner et al. 2021 "On estimating the shape and dynamics of phenotypic distributions in ecology and evolution", where they present a bootstrapping method that allows including ITV without measuring everything in the field).

Other minor concern is that they do not account for other important drivers of trait variation at the local scale, such as edaphic factors or altitude (which can be related to soil variation too), potential evapotranspiration, land-use, legacy effect...they also acknowledge these issues in the discussion. Given

the geographic extent of the study, how they increase the complexity of the models with increasing the narrowness of habitat definition, and the solely focus on the effects climate I don't consider this is a major issue.

Beyond these minor issues, I think is important and would appreciate to know whether the authors have check for normality in the distribution of the models' residuals, and how the residuals distribution look like.

Reviewer #1:

Kambach and colleagues tested whether community weighted means (CWMs) in four functional traits (seed mass, plant height, specific leaf area (SLA), specific root length (SRL)) are related to climate variables and whether the explanatory power of the climate variables get stronger if one accounts for habitat type. To this aim, they combined more than 300,000 vegetation records from the European Vegetation Archive, which have been assigned to the hierarchy of EUNIS habitat types, with climatic data from the Chelsa database and with gap-filled trait data from the TRY database. They found that for CWMs of SLA and SRL, the variance explained by climatic variables increased when they used finer habitat categories. They also found that the relationship between CWMs and climate may depend on the habitat type, and they therefore conclude that it is important to account for habitat type.

The manuscript is overall clearly written, but as it is very brief, I am missing some important details on the methods and I am also missing some results or discussion thereof. For example, although for CWMs of SLA and SRL, the amount of variance explained by climate increases when they included finer and finer habitat categories, it is not mentioned that for three of the traits, the explained variance in CWMs is actually highest when habitat is not considered. This makes sense if one considers that habitat types are also determined by climate. It would be useful to discuss this and also to show the positions of the habitat types in PCA plots (or tables).

Response: We completely agree with the reviewer here and have added this observation of maximized explained variation by climate in L 191-193 and discussed (and show) the implication of the direct climate effect on habitat type distribution in L 220-223 and Supplementary Fig. S1-S4.

As the introduction nicely starts with functional traits and main axes of variation, such as the fast-slow continuum, it would be nice if they could point out how the four chosen traits relate to these axes. For example, based on a recent paper by Carmona et al. ¹, plant height and seed mass would both represent their first axis, SLA their second axis, and SRL their third axis. So, the traits chosen represent three of the major axes of trait variation, and this is a strength of this study.

Response: We agree with the reviewer and thus added in the description of the four analysed traits that “Previous research has shown that these four traits can capture the major gradients in the global spectrum of plant form and function (plant height, SLA and seed mass, ²) and the roots economic spectrum (SRL, ^{1,3}).” (L 144-147). We furthermore now provide the position of the four analysed traits along the first two principal components of together with 24 additional gap-filled traits from the TRY plant trait database (Supplementary Fig. S6 and L 359-361).

Of the more than 1.7 million vegetation plots in the European Vegetation Archive, the authors only used the c. 300,000 that had trait data for the species that covered more than 80% of the plots. This is good. However, as the authors used the gap-filled (imputed) trait data of TRY, I wonder how good some of the trait data and thus the values of the CWMs are. For details on the gap-filling, one is referred to another paper, but the authors should mention for how many of the 6,586 species used in the analyses, the data was imputed. In the Carmona et al. study on the global spectrum of plant traits and function, they has SRL data for only 1,736 species. I assume that this number is not higher here, for the European flora, which would mean that

measured SRL data would only be available for up to 25% of the species and that the remaining data for SRL have been imputed. This does not mean that the data are not correct, but it would be good to provide the details. If there is large variation in the percentage species/coverage in CWMs that are based on 'real' and imputed data, one could consider using the percentage of real data per plot as a weighting factor in the analyses.

Response: We acknowledge that we used imputed trait values for a significant proportion of species. In order to make this fact more transparent, we now added a section in the methods where we reported the proportion of species with available trait data and the average plot-level cover of species with available trait data: "Without the imputed values, trait observations were available for 45% (plant height), 28.4% (SLA), 34.4% (seed mass) and 5.4% (SRL) of all species in the dataset used for analysis, which accounted for an average of 95.6%, 91.4%, 90.8% and 47.4% of the plot-level plant cover, respectively." (L 363-366).

Since the species that had actual trait data contributed, on average, a significant proportion of the plot-level cover values, we do not see the need to apply a weighting scheme which might also introduce bias because different habitat types might show a different proportion of available plant trait data.

I would also like to see more details on the actual statistical analyses. The authors used generalized additive models (GAMs). As they also included random terms, I assume that it were generalized additive mixed models (GAMMs). I am not a specialist in GAM(M)s, but I thought that they fitted non-linear functions to the data, but based on the figures, it seems that only linear relationships were fitted. So, how do the used models differ from generalized linear (mixed) models? Also, the method of calculating the marginal R² is based on the paper by Nakagawa and Schielzeth (Ref. 56), which is about GLMMs. Again, this does not mean that the analyses are wrong, but more details on the analyses should be provided.

Response: The main motivation to use GAMMs instead of GLMMs was to include a spline-on-the-sphere term that allowed to account for spatial autocorrelation. Otherwise, the GAMMs included only linear terms as in ordinary linear regressions. In consequence, observations that were close to each other were weighted less than those more distantly apart. As suggested, we included more details on the applied models and their implementation in R. We listed all the model terms for each model (L 399-414), provide the model code on GitHub and show the model residuals in Supplementary Fig. S9-S13 (L 413-414).

Minor comments

Line 156: place 'the' before 'more'

Response: We adapted the suggested grammatical improvement.

Line 168: CWMs, not CMWs

Response: We corrected this mistake.

Line 170: insert 'also' before 'associated' to indicate that it is the same pattern as for the Mediterranean gradient.

Response: We adapted the suggested grammatical improvement.

Line 250-252: Here the authors could expand a bit on why it is likely that plant height and seed mass might be more strongly driven by habitat. To me this seems obvious given that plant height is strongly associated with woodiness, and that therefore e.g. forest habitats have much higher plant heights than grasslands. And seed mass is frequently associated with plant height (both align well along trait-spectrum axis 1 of Carmona et al. 2021)

Response: We now state clearly that plant height is strongly determined by the occurrence of woody species which might be determined by local land use and soil conditions rather than climatic gradients. “The two traits that did not follow this pattern, i.e. community plant height and seed mass, are both strongly related along the global spectrum of plant form and function^{1,2} and most strongly determined by the habitat type. Community plant height relates to the occurrence of woody species, which markedly depends on land use and is furthermore more strongly determined by soil moisture and nutrients than broad climatic conditions⁴. (L 259-264).

Line 288: The focus on linear effects is mentioned as a limitation. I do not see this as a major limitation, but when the authors used GAMs, I wonder why they did not allow them to be non-linear.

Response: We were generally interested in the linear effects of climate on plant traits, and we applied GAMMs only to be able to correctly account for the spatial dependency between plots. Including non-linear effects would have shifted the focus of the study towards linear versus non-linear effects and would have hampered any direct comparison of the direction and strength of climate-trait relationships between different habitat types. Please note that we also did not account for interactive effects of drivers, which might also result in non-linear relationships. This is an interesting topic, which however, would have been beyond the scope of our paper.

Line 302: Add here that another limitation of the data are that many data were imputed.

Response: As suggested, we now clearly report the proportion of species with available trait data and the average plot-level cover of species with available trait data: “Without the imputed values, trait observations were available for 45% (plant height), 28.4% (SLA), 34.4% (seed mass) and 5.4% (SRL) of all species in the dataset used for analysis, which accounted for an average of 95.6%, 91.4%, 90.8% and 47.4% of the plot-level plant cover, respectively.” (L 363-366).

Line 325-326: I assume that these latitude cut-offs correspond to the most southern and northern latitudes of Europe, but it would be good to explicitly mention why these cut-offs were chosen.

Response: We specified the application of latitudinal cut-offs as “we removed all plots... with latitudinal coordinates lower than the southernmost point of 34° or higher than the northernmost point of 82°” (L 335-336).

Line 330: Add the approximate size in km² of the 0.01° grid cells.

Response: We added the approximated size of the grid cells as “approximately 1.11 x 0.69 km in Central Europe” (L. 341).

Line 347-348: Provide more information on the gap-filling. For how many species per trait was this done, and how good were the gap-filling statistics?

Response: We now clearly state the proportion of species with available (i.e. non-imputed) trait data together with the average plot-level cover of these species with available trait data: “Without the imputed values, trait observations were available for 45% (plant height), 28.4% (SLA), 34.4% (seed mass) and 5.4% (SRL) of all species in the dataset used for analysis, which accounted for an average of 95.6%, 91.4%, 90.8% and 47.4% of the plot-level plant cover, respectively.” (L 363-366).

Line 388: Here and elsewhere, the habitat categories start with a capital letter. Initially, this confused me, but then I realized that it is most likely the code for the habitat type. To avoid confusion, I recommend to put delete the code from the text or to put it in brackets after the written-out name of the habitat type: sparsely vegetated habitats (U).

Response: We adapted the suggested naming of habitat types with the habitat code in following in parentheses throughout the whole manuscript.

Fig. 1: I assume that the range of x-axis values covered by the lines varies because some of the habitat types did not cover the full range of principal component scores. It would be good to mention this in the caption. It would also be helpful to plot the average or median principal component score of each habitat type, either as a dot on the corresponding line, or as vertical stripes on the x-axis.

Response: We now provide the distribution of the broad level 1 and intermediate level 2 habitat types along the four climatic principal components in Supplementary Fig. S1-S4 (L 174-175).

Reviewer #2:

Habitat-specificity of climate-trait relationships

Stephan Kambach and colleagues evaluated relationships between climate and plant functional traits across fine scales in Europe. By calculating community-weighted means of plant height, SLA, seed mass and SRL within vegetation plots, they found increases in predictive power of climate varying from broader to narrow finer scales for SLA and SRL but not height and seed mass. The findings have implications for understanding the drivers of plant distribution especially for predicting the effects of climate on species distributions.

Because of the very narrow focus of the work, I recommend that it may be more suited for a discipline-focused outlet dealing with plant ecology than a general-readership journal like Nature Communications as I find the 'general readership' component, i.e., why a general readership should care, missing. However, I leave this to the editor to decide.

I have the following comments below:

1) It was not clear why the authors focused only on the four plant functional traits: plant height, SLA, seed mass and SRL. The less traits data may give unreal signature, or cannot represent the full nature of climate-trait relationships.

Response: The reason that we focused our analysis on these four functional traits is that they already capture a large proportion of trait variation in the aboveground global spectrum of plant form and function as well as the belowground roots economics spectrum¹ (see species-level trait PCA in S6 and L143-146). Thus, with only these four traits, we were able to capture the main gradients in species life-history strategies and their relationships with climatic gradients instead of “fishing” for the statistically most significant climate-trait relationships. Supplementarily, we now also provide the results of a principal components analysis to show the relationships of the four traits with 24 additional traits from the gap-filled data of the TRY plant trait database (Supplementary Fig. S6, L 359-361).

2) In the context of this analysis, functional traits are properties of species. However, species are not independent units--instead, they are descended from shared phylogenetic ancestry. Yet, I could not find anywhere the authors accounted for such phylogenetic autocorrelation or even acknowledge this fact. Such autocorrelation can surely influence the results presented here.

Response: Please note that we did not carry out a species-level analysis (except for a newly included PCA to illustrate the species-level trait interrelationships) but analyzed the functional composition at the community level. Thus, phylogenetic correlation between species is not an issue. Instead, phylogenetic relationships among species might result in phylogenetic overdispersion or phylogenetic clustering in communities. Under a model of evolutionary trait conservatism, these patterns might result in trait (=phenotypic) overdispersion or clustering. However, analyzing these patterns would have required a completely different approach. Most importantly, in this approach a definition of the species pool for a given habitat type would have to be made, for which we do not see an obvious solution at the European scale. In this context, we aggregated the trait data at the level of communities and not at the species-level, phylogenetic similarity or dissimilarity can be considered an intrinsic property of habitat types that share a floristic composition and evolutionary history.

3) While the study dwelled on the scale-dependence of habitat and climate on plant functional traits, it was not clear the size of the different habitat types examined. For instance, the authors used ambiguous terms like 'narrower' or 'broad' habitats, but it is not clear for a reader how big or small these habitats actually are. In Line

Response: We apologize for this misunderstanding. The terms broad and narrow refer to the hierarchy of habitat types. Spatial scale-dependence is not addressed in this study. To make this clearer we slightly rephrased the description of the habitat classification system as: “We used the first three levels of the classification hierarchy, ranging from the highest level 1 (broadest habitat types, e.g., forests – habitat code T) through the intermediate level 2 (narrower types, e.g., broadleaved deciduous forests – T1) to the lowest level 3 (narrowest types, e.g., temperate Salix and Populus riparian forest – T11).” (L 371-375).

Reviewer #3:

The authors explore climate-trait relationships to test a foundational assumption of trait-based ecology: that “the geographical distribution of dominant functional traits in plant communities is shaped by macro-environmental gradients, independently of taxonomy”. That is to say, the authors want to test whether there is some generality in the trait values geographical distribution according to macro-environmental gradients.

They argue that trait-climate relationships at a global scale are often weak or inconsistent because global climate-traits relationships might be blurred or masked by local conditions, habitat-specific factors, or different evolutionary histories. To understand whether climate-trait relationships are global or habitat specific, and whether these relationships are invariant to habitat narrowness definition, the authors suggest restricting the analysis to smaller units (more homogenous units) like for example habitats that shared floristic composition, or evolutionary history, that reflect common adaptations that are probably independent of global climate gradients. So, they can control these confounding or interacting factors. Within these habitats they also suggest analyzing climate-trait relationships using different levels of habitat definition by changing the scale, basically (if I am understanding properly what they called “habitat narrowness”). This understanding might improve trait-climate predictions and help to understand the consequences of functional variation on ecosystem functioning. To answer these questions, the authors employed an impressive vegetation dataset from the European Vegetation Archive, 19 high-resolution CHELSA bioclimatic variables, and they characterized the CWM of four functional traits obtained from TRY plant trait database. The habitat types were classified according to the EUNIS classification, with three levels of narrowness (broad-intermediate-narrow habitats).

They found that climate is a significant predictor of CWMs of all plant traits analyzed in all habitats, which is in line with previous studies. They find that decomposing the analysis into narrower habitats improve climate-trait relationships for some traits (SLA and SRL), but not all. The authors found that trait composition was largely determined by habitat type (that is to say, there are strong habitat-specificity in the functional composition of communities). However, they were able to demonstrate a strong effect of climate on functional composition of communities irrespective of habitat, which from my point of view is their most noteworthy result.

General comments to the study:

I find the work original, well written, easy to follow, the methods are state of art to me, and it also provide evidence on climate-traits relationships. In addition, this work contributes to test a foundational assumption of trait-based ecology, that has been relatively poorly tested and that is key for understanding climate impacts on communities, and ecosystems. I personally appreciate to see this kind of works in the literature since they help to solidify foundational trait-based ecology postulates. I have read the work carefully and did not find any major issues to point out. I would just appreciate to see the distribution of the models’ residuals to check whether the model is correctly specified.

Response: We now added an overview on the distribution of the residuals for each model in Supplementary Fig. S9-S13 (L 413-414).

Regarding details for reproducibility. I consider the authors provided enough details about the

data employed and the methods. However, a reproducible code for harmonization, data cleaning and data analysis is missing. Also, the final dataset is missing (or at least I couldn't find them). In my opinion, at least the data analysis codes and final data set should be available for the readers in a final version of this manuscript.

Response: We now provide the full code for the analyses and figures on GitHub (L 441-445), and we stored the dataset on the plot coordinates, community-weighted means and first four principal components from the 19 bioclimatic variables in Dryad (432-439). Raw bioclimatic variables can be downloaded at chelsa-climate.org/bioclim and raw plant trait data can be requested at www.try-db.org/TryWeb/dp.php." (L 441-445).

Minor comments on the methods and data analysis:

I do not find any major issues or flaws in the methods and data analysis. I found the data analysis appropriate to test their hypothesis and the work support the conclusions and claims. Although already addressed in the text, I would like to point out a few minor concerns.

First, the authors calculate CWM from species-level trait averages. They do not use in-situ/local measurements neither any measure of intraspecific variation in their analysis. They acknowledge this shortcoming and justify the decision due to the large geographic extent of the study what is reasonable to me. Although not needed for this study, it would be awesome to have local measures, or some proxy of the intraspecific trait variation (ITV) in the field to see if there are improvements on climate-trait relationships (Although is still a preprint you can have a look to Maitner et al. 2021 "On estimating the shape and dynamics of phenotypic distributions in ecology and evolution", where they present a bootstrapping method that allows including ITV without measuring everything in the field).

Response: We agree that it would be great to be able to analyse intraspecific trait variability along these climatic gradients but, unfortunately, we do not have such data.

Other minor concern is that they do not account for other important drivers of trait variation at the local scale, such as edaphic factors or altitude (which can be related to soil variation too), potential evapotranspiration, land-use, legacy effect...they also acknowledge these issues in the discussion. Given the geographic extent of the study, how they increase the complexity of the models with increasing the narrowness of habitat definition, and the solely focus on the effects climate I don't consider this is a major issue.

Response: It is correct, that we did not account for several potentially important drivers of plant trait distribution but, as noted by the reviewer, the general idea behind applying the habitat types was to account for a significant proportion of variability in those otherwise unaccounted local factors for which we are otherwise lacking the necessary data.

Beyond these minor issues, I think is important and would appreciate to know whether the authors have check for normality in the distribution of the models' residuals, and how the residuals distribution look like.

Response: We visually inspected and now provide all model residuals in Supplementary Fig. S9-S13 (L 413-414).

References

1. Carmona, C. P. *et al.* Fine-root traits in the global spectrum of plant form and function. *Nature* **597**, 683–687; 10.1038/s41586-021-03871-y (2021).
2. Díaz, S. *et al.* The global spectrum of plant form and function. *Nature* **529**, 167–171; 10.1038/nature16489 (2016).
3. Bergmann, J. *et al.* The fungal collaboration gradient dominates the root economics space in plants. *Science advances* **6**; 10.1126/sciadv.aba3756 (2020).
4. Ding, J., Travers, S. K. & Eldridge, D. J. Occurrence of Australian woody species is driven by soil moisture and available phosphorus across a climatic gradient. *J Veg Sci* **32**, e13095; 10.1111/jvs.13095 (2021).

REVIEWER COMMENTS

Reviewer #1 (Remarks to the Author):

Overall, I am happy with the revisions and the responses to my previous comments. However, I think that it is a pity that the authors did not follow my suggestion of using the percentage of real data per plot as a weighting factor in the analyses. Given that the species with real height, SLA and seed mass data accounted on average for over 90% of the plant cover in the plots, I believe that these results will be robust. However, for SRL, the species with real data accounted on average for less than 50% of the plant cover in the plots. So, at least for the latter trait it would be interesting to see how robust the findings are. The authors responded to my comment by writing: "...we do not see the need to apply a weighting scheme which might also introduce bias because different habitat types might show a different proportion of available plant trait data.". I do not follow this argument, because even if habitat types differ in this regard, I do not see how this would affect the trait-climate relationships within the habitat types.

L224-225: Replace "for some traits (SLA and SRL) but not for all" with "for SLA and SRL but not for height and seed mass".

L255: I assume "mycorrhizal fungi" are meant, not "mycorrhizal bacteria". It would also be good to mention somewhere explicitly that low SRL (thick roots) is typical for species that have mycorrhizal associations.

L356-367: The way it is currently written, it is not entirely clear that gap filling (imputation) refers to the species without species- or genus-level average trait values (some readers might think that the use of genus-level average trait values was the gap filling). So, mention explicitly that for the species without species- and genus-level average trait data, data imputation was used. It would also be good to indicate a few specifics about the imputation method (Was it based on phylogenetic relationships among species? Do you have some statistics on how good the imputation was?)

Supplementary Fig. S9-13: The residual plots for all 4 traits look exactly the same. Please, check.

Reviewer #1:

Overall, I am happy with the revisions and the responses to my previous comments.

Response: Thank you for your patience and positive attitude towards our study. We really appreciate your comments and feel that they have significantly improved our MS.

However, I think that it is a pity that the authors did not follow my suggestion of using the percentage of real data per plot as a weighting factor in the analyses. Given that the species with real height, SLA and seed mass data accounted on average for over 90% of the plant cover in the plots, I believe that these results will be robust. However, for SRL, the species with real data accounted on average for less than 50% of the plant cover in the plots. So, at least for the latter trait it would be interesting to see how robust the findings are. The authors responded to my comment by writing: "...we do not see the need to apply a weighting scheme which might also introduce bias because different habitat types might show a different proportion of available plant trait data.". I do not follow this argument, because even if habitat types differ in this regard, I do not see how this would affect the trait-climate relationships within the habitat types.

Response: In order to accommodate this legitimate concern of the validity of our results with regard to missing trait data, we now repeated all statistical analyses, and (as suggested by the reviewer) weighted the impact of each plot according to the availability of measured trait data: *"To check if the obtained results depend on the proportion of species with actual trait measurements, we repeated all analyses with models for which the contribution of each plot was weighted according to the summed cover of species with trait measures divided by the total plant cover. The results of these weighted analyses were qualitatively similar to the results from unweighted analyses and shown in Fig. S21-S25" (L 430-435).*

L224-225: Replace "for some traits (SLA and SRL) but not for all" with "for SLA and SRL but not for height and seed mass".

Response: We changed the wording as suggested. (L 225-226)

L255: I assume "mycorrhizal fungi" are meant, not "mycorrhizal bacteria". It would also be good to mention somewhere explicitly that low SRL (thick roots) is typical for species that have mycorrhizal associations.

Response: Indeed, that was a mistake from our side. We corrected this to "mycorrhizal fungi". We now also mention that thicker roots are an indicator for mycorrhizal associations in the introductory Table 1: *"Central to species' collaborative strategy as species with mycorrhizal associations tend to have lower SRL (thicker roots)."* (L 654)

L356-367: The way it is currently written, it is not entirely clear that gap filling (imputation) refers to the species without species- or genus-level average trait values (some readers might think that the use of genus-level average trait values was the gap filling). So, mention explicitly that for the species without species- and genus-level average trait data, data imputation was used. It would also be good to indicate a few specifics about the imputation method (Was it

based on phylogenetic relationships among species? Do you have some statistics on how good the imputation was?)

Response: We rephrased this section to make it more clear for the reader that we requested the (already) gap-filled data from the TRY plant trait database (for which empty species-trait combinations were filled by an algorithm that used information on the TRY plant taxonomy). Only in the cases where we could not match our species to the gap-filled TRY data, we calculated genus-level average trait values. We now added plots on the predicted versus measured trait values together with the root-mean-square error (Fig. S14): *“We extracted species- and genus-level average values for plant height, SLA, seed mass and SRL from the TRY plant trait database³⁰ version 5 which covered species-level mean values of 33 traits for 50,404 species, derived with Bayesian Hierarchical Probabilistic Matrix Factorization. This algorithm predicts trait values at the level of individuals based on the observed trait records, the observed trait-trait correlations, and the taxonomic hierarchy reflecting phylogeny^{54,55}. Root-mean-square error of z-transformed predicted versus available trait values ranged from 0.07 (seed mass) to 0.2 (SLA) (see Fig. S14). From this gap-filled dataset, we assigned species-level mean trait values to 6,025 species from our dataset and for an additional 690 species, we assigned genus-level mean trait values from the gap-filled dataset.”* (L356-365).

Supplementary Fig. S9-13: The residual plots for all 4 traits look exactly the same. Please, check.

Response: We are very grateful for this remark because this was really a bug (of exactly one character) in our code for automatic figure construction. As a consequence, we re-checked our entire code and updated Fig. S16-S20 with the correct residual plots. All other result were not affected by this bug.